# Gradient-based meta-solving and its applications to iterative methods for solving differential equations

## Abstract

In science and engineering applications, it is often required to solve similar computational problems repeatedly. In such cases, we can utilize the data from previously solved problem instances to improve efficiency of finding subsequent solutions. This offers a unique opportunity to combine machine learning (in particular, meta-learning) and scientific computing. To date, a variety of such domain-specific methods have been proposed in the literature, but a generic approach for designing these methods remains under-explored. In this paper, we tackle this issue by formulating a general framework to describe these problems, and propose a gradient-based algorithm to solve them in a unified way. As an illustration of this approach, we study the adaptive generation of initial guesses for iterative solvers to speed up the solution of differential equations. We demonstrate the performance and versatility of our method through theoretical analysis and numerical experiments.

## 1 Introduction

It is common and important in science and engineering to solve similar problems repeatedly. For example, in material science, a tremendous amount of physical and numerical experiments are conducted to discover and characterize new materials (Schmidt et al. (2019)). For another example, in computational fluid dynamics, many methods involve solving a Poisson equation to compute the pressure field in every time step of the simulation (Ajuria Illarramendi et al. (2020)). In these situations, we can utilize the data of the previously solved problems to solve the next similar but unseen problems more efficiently, and machine learning is a natural and effective approach for this.

Thus, in recent years, many learning-based methods have been proposed for repeated solutions of computational problems such as partial differential equations (PDEs) (Tang et al. (2017); Özbay et al. (2021); Tompson et al. (2017); Hsieh et al. (2018); Huang et al. (2020)). For example, in Tang et al. (2017), Ajuria Illarramendi et al. (2020), and Özbay et al. (2021), convolutional neural networks are used to predict the solution of Poisson equations, and Ajuria Illarramendi et al. (2020) and Özbay et al. (2021) propose to use the predicted solution as an initial guess of traditional numerical methods. Hsieh et al. (2018) combines a neural network and an iterative solver to accelerate it with maintaining the convergence guarantee. In addition to the regular supervised learning approaches, there are several works where meta-learning approaches are taken to solve computational problems (Feliu-Fabà et al. (2020); Chen et al. (2020); Psaros et al. (2021); Guo et al. (2021)). Meta-learning, or learning to learn, leverages previous learning experiences to improve future learning performance (Hospedales et al. (2021)), which fits the motivation utilizing the data from previously solved equations for the next one. For example, Chen et al. (2020) use meta-learning to generate a smoother of the Multi-grid Network for parametrized PDEs. Psaros et al. (2021) propose a meta-learning technique for offline discovery of physics-informed neural network loss functions.

Although many methods have been proposed in this direction, they are often problem-specific. In other words, there lacks both a unified framework to explain them and a general design pattern to adapt them to new problem settings. On the other hand, in the machine learning literature, there exists a general methodology - gradient-based meta-learning (Finn et al. (2017)) - that covers a variety of meta-learning problem settings. In this paper, we generalize this approach to yield a framework,

which we call gradient-based meta-solving (GBMS), that encompasses both learning and computational problems. This offers a general means to understand and develop learning-based algorithms to speed up computation. As an illustration of our approach, we apply GBMS to accelerate the solution of differential equations with iterative methods through learning. We show the advantage of the proposed algorithm over the baseline classical and learning-based approaches through theoretical analysis and numerical experiments. Finally, we incorporate the algorithm into a practical application and demonstrate its versatility and performance.

## 2 GRADIENT-BASED META-SOLVING

In this section, we introduce our core idea of gradient-based meta-solving. First, we formulate a class of problems, which we call meta-solving, that includes both ordinary meta-learning and learning-based computational problems. We then propose a general gradient-based algorithm for solving them. Under our formulation, many methods proposed in related works can be regarded as special cases of the GBMS algorithm.

### 2.1 GENERAL FORMULATION OF META-SOLVING

Let us now introduce the general formulation of meta-solving. We first fix the required notations. A task $\tau$ is a tuple $\tau = (D_\tau, \mathcal{U}_\tau, L_\tau)$ consisting of a dataset $D_\tau$, a solution space $\mathcal{U}_\tau$, and a loss function $L_\tau$. A solution space $\mathcal{U}_\tau$ is a set of parametric candidate solutions, which is usually a subset of $\mathbb{R}^N$ for some $N \in \mathbb{N}$. A loss function $L_\tau$ is a function from $\mathcal{U}_\tau$ to $\mathbb{R}_{\geq 0}$ that measures the quality of solution candidates. To solve a task $\tau$ means to find an approximate solution $\hat{u} \in \mathcal{U}_\tau$ which minimizes $L_\tau(\cdot)$. Meta-solving considers the solution of not one, but a distribution of tasks by a learnable solver. Thus, we consider a task space $(\mathcal{T}, P)$ as a probability space that consists of a set of tasks $\mathcal{T}$ and a task distribution $P$, which is a probability measure defined on a suitable $\sigma$-algebra on $\mathcal{T}$. A solver $\Phi$ is a function from $\mathcal{T} \times \Theta$ to $\mathcal{U}$, where $\mathcal{U} = \bigcup_{\tau \in \mathcal{T}} \mathcal{U}_\tau$. $\theta \in \Theta$ is the parameter of $\Phi$, and $\Theta$ is its parameter space. Here, $\theta$ may or may not be trainable, depending on the problem. Then, solving a task $\tau \in \mathcal{T}$ by an algorithm $\Phi$ with a parameter $\theta$ is denoted by $\Phi(\tau; \theta) = \hat{u}$. A meta-solver $\Psi$ is a function from $\mathcal{T} \times \Omega$ to $\Theta$, where $\omega \in \Omega$ is a parameter of $\Psi$ and $\Omega$ is its parameter space. A meta-solver $\Psi$ parametrized by $\omega \in \Omega$ is expected to generate an appropriate parameter $\theta_\tau \in \Theta$ for solving a task $\tau \in \mathcal{T}$ with a solver $\Phi$, which is denoted by $\Psi(\tau; \omega) = \theta_\tau$. Then, by using the notations above, our meta-solving problem is defined as follows:

**Definition 1** (Meta-solving problem). *For a given task space $(\mathcal{T}, P)$, solver $\Phi$, and meta-solver $\Psi$, find $\omega \in \Omega$ which minimizes $\mathbb{E}_{\tau \sim P}[L_\tau(\Phi(\tau; \Psi(\tau; \omega)))]$.*

We present some familiar examples, which can be regarded as special cases of the meta-solving problem. First, we can see that conventional meta-learning problems fall in this formulation.

**Example 1** (Few-shot learning). As an example of the meta-learning problem, we take the few-shot regression problem with MAML (Finn et al. (2017)). The components of the problem are the following. The task $\tau$ is to learn a regression model from data. The dataset is $D_\tau = \{(x_i, y_i)_{i=1}^K\}$, which satisfies $y = f_\tau(x)$ for an unknown function $f_\tau$. $D_\tau$ is divided into the training set $D_\tau^{\text{train}}$ and validation set $D_\tau^{\text{val}}$. The solution parameter space $\mathcal{U}_\tau$ is a weights space of a neural network (NN) that models $f_\tau$. Note that $\mathcal{U} = \mathcal{U}_\tau$ because the architecture of NN is shared across all tasks. The approximate solution $\hat{u} \in \mathcal{U}$ is the trained weights of NN, which is obtained by training on $D_\tau^{\text{train}}$. The loss function $L_\tau : \mathcal{U} \to \mathbb{R}_{\geq 0}$ is the mean squared error (MSE) on $D_\tau^{\text{val}}$. The task distribution $(\mathcal{T}, P)$ is determined by the distribution of the target function $f_\tau$ and distribution of samples $(x, y)$. The solver $\Phi : \mathcal{T} \times \Theta \to \mathcal{U}$ is the single step gradient descent to minimize $L_\tau^{\text{train}}(u) = \frac{1}{|D_\tau^{\text{train}}|} \sum_{(x,y) \in D_\tau^{\text{train}}} \|\text{NN}(x; u) - y\|^2$. Its parameter $\theta \in \Theta$ is initial weights $u^{(0)}$ of NN, so $\Theta = \mathcal{U}$. Thus, $\Phi(\tau; \theta) = u^{(0)} - \alpha \nabla_{u^{(0)}} L_\tau^{\text{train}}(u^{(0)}) = \hat{u}$, where $\alpha$ is a learning rate. The meta-solver $\Psi : \mathcal{T} \times \Omega \to \Theta$ is considered as a constant function that returns its parameter $\omega \in \Omega$ for any task $\tau \in \mathcal{T}$. The parameter $\omega \in \Omega$ is expected to be an appropriate initial weights for all $\tau \in \mathcal{T}$ to be fine-tuned easily. Thus, $\Omega = \Theta = \mathcal{U}$, and $\Psi(\tau; \omega) = \omega = \theta = u^{(0)}$. Note that the output of the meta-solver, the initial weights $u^{(0)}$, does not depend on $\tau$. Then, the few-shot learning problem is defined as meta-solving problem, which is

$$\min_\omega \mathbb{E}_{\tau \sim P}[L_\tau(\Phi(\tau; \Psi(\tau; \omega)))] = \min_\omega \mathbb{E}_{\tau \sim P} \sum_{(x,y) \in D_\tau^{\text{val}}} \|\text{NN}(x; \omega - \alpha \nabla_\omega L_\tau^{\text{train}}(\omega)) - y\|^2. \quad (1)$$

In addition to the conventional learning problems, we can regard other computational problems, such as solving a differential equation, as a task of the meta-solving problem. In constrast with MAML, the inner-loop learning is now replaced with a family of iterative solvers for differential equations. This necessitates the distinction of the meta-solver parameter space $\Omega$ and the solution space $\mathcal{U}$. Moreover, the meta-solver has to produce a task-specific parameter for the inner solver.

**Example 2** (Solving differential equations). Suppose that we need to repeatedly solve similar instances of a class of differential equations with a given numerical solver. The solver has a number of hyper-parameters, which sensitively affect accuracy and efficiency depending on the problem instance. Thus, finding a strategy to optimally select solver hyper-parameters given a problem instance can be viewed as a meta-solving problem. The components of the problem are the following. The task $\tau$ is to solve a differential equation. In this example, suppose that the target equation is the Poisson equation $-\Delta u = f_\tau$ with Dirichlet boundary conditions $u = g_\tau$. The dataset $D_\tau$ contains data of the differential equation, $D_\tau = \{f_\tau, g_\tau\}$. The solution parameter space $\mathcal{U}_\tau$ is $\mathbb{R}^{N_\tau}$. The loss function $L_\tau : \mathcal{U}_\tau \to \mathbb{R}_{\geq 0}$ measures the accuracy of $\hat{u} \in \mathcal{U}_\tau$. In this example, the $\ell^2$-norm of the residuals obtained by substituting the approximate solution $\hat{u}$ into the equation can be used. The task distribution $(\mathcal{T}, P)$ is the joint distribution of $f_\tau$ and $g_\tau$. The solver $\Phi : \mathcal{T} \times \Theta \to \mathcal{U}$ is a numerical solver with a parameter $\theta \in \Theta$ for the differential equation. In this example, suppose that $\Phi$ is the Jacobi method (Saad (2003)) and $\theta$ is its initial guess. The meta-solver $\Psi : \mathcal{T} \times \Omega \to \Theta$ is a strategy characterized by $\omega \in \Omega$ to select a parameter, an initial guess, $\theta_\tau \in \Theta$ for each task $\tau \in \mathcal{T}$. Note that the output of the meta-solver depends on $\tau$, which is different from the case of Example 1. Then, finding the strategy to select parameters of the numerical solver becomes meta-solving problem.

The above examples explain why we describe our problem as meta-solving instead of meta-learning. In Example 1, the task $\tau$ is learning from data, and the solver $\Phi$ is gradient descent. In Example 2, the task $\tau$ is solving a differential equation, and the solver $\Phi$ is an iterative differential equation solver. Regardless of the type of task and algorithm, in both cases, the solver $\Phi$ solves the task $\tau$, and we do not distinguish whether $\Phi$ is a learning algorithm or other numerical solver. In other words, learning algorithms such as gradient descent are also a type of numerical solvers, and we regard learning as a special case of solving. It is also true for the outer learning algorithm to learn how to solve the task $\tau$ with the solver $\Phi$, so learning to solve is a special case of solving to solve. In this sense, we call it meta-solving.

## 2.2 GRADIENT-BASED META-SOLVING

In the previous section, we defined meta-solving problems as a generalization of meta-learning problems, but how can we solve them effectively? For meta-learning problems such as Example 1, general methodologies in the form of gradient-based meta-learning (Hospedales et al. (2021)), e.g. MAML (Finn et al. (2017)), have been proposed. In Example 1, MAML solves the minimization problem (1) by updating $\omega$ using the outer gradient descent algorithm $\omega \leftarrow \omega - \beta \nabla_\omega L_\tau(\hat{u})$. MAML is an algorithm to find good initial weights of the neural network for conventional meta-learning problems, but it can be generalized to the meta-solving problems. In meta-solving problems, tasks may not be learning problems, $\Phi$ may not be gradient descent, and its parameter $\theta$ may not be initial weights of a neural network. However, we can still employ the same update rule as MAML, as long as $L_\tau$, $\Phi$, and $\Psi$ are differentiable. Thus, for differentiable solvers $\Phi$ and $\Psi$, we propose gradient-based meta-solving algorithm (Algorithm 1) as a generalization of MAML. This is a form of data-driven algorithm design, and differs from previous works in this direction (e.g. Hutter et al. (2011), Mitzenmacher & Vassilvitskii (2021), and Balcan (2021)). These focus on discontinuous problems such as combinatorial optimization, while our framework focuses on differentiable problems.

## 2.3 ORGANIZING RELATED WORKS

Owing to the general formulation presented above, we can organize several related works on learning-based methods for scientific computing and describe them in a unified way. We highlight the advantages of our method using the examples in this section. The detailed description of how the task, the solver and the meta-solver are defined in each case are found in Appendix A.

First, let us review Feliu-Fabà et al. (2020). This work proposes a neural network architecture inspired by the nonstantard wavelet form with meta-learning approach to solve the equations containing partial differential or integral operators. It can be regarded as a special case of GBMS, where

---

**Algorithm 1:** Gradient-based meta-solving algorithm

---

**Require:** $(P, \mathcal{T})$: task space, $\Phi, \Psi$: differentiable solver, $S$: stopping criterion, $\alpha$: learning rate
**Result:** $\omega$

1 initialize $\omega$;
2 **while** *S is not satisfied* **do**
3      Sample task $\tau \sim P$;
4      Generate parameter of $\Phi$ using meta-solver: $\theta_\tau(\omega) \leftarrow \Psi(\tau; \omega)$;
5      Solve task $\tau$ using solver: $\hat{u}(\omega) \leftarrow \Phi(\tau; \theta_\tau(\omega))$;
6      Update meta-solver parameter: $\omega \leftarrow \omega - \alpha \nabla_\omega L_\tau(\hat{u}(\omega))$;
7 **end**

---

task $\tau$ is solving the equation, solver $\Phi$ is the forward computation of the trained neural network, $\theta \in \Theta$ is its weights, and meta-solver $\Psi$ is the constant function that returns the weights $\theta = \omega \in \Omega$. Note that $\theta$ does not depend on the task $\tau$. We also note that other works where neural networks replace a whole or part of numerical methods (Tang et al. (2017); Özbay et al. (2021); Tompson et al. (2017); Hsieh et al. (2018)) can be organized in the same way. Thus, the meta-solving formulation includes many learning-based methods where meta-learning techniques are not explicitly employed.

We take Psaros et al. (2021) as another example. In this work, meta-learning is used to learn a loss function of the physics-informed neural network, shortly PINN (Raissi et al. (2019)), for solving PDEs. This also can be considered as a special case of GBMS, where $\tau$ is training the PINN, the solver $\Phi$ is the gradient descent for training the PINN, $\theta \in \Theta$ is the weights of another neural network used as the loss function in the training, and the meta-solver $\Psi$ is the constant function that returns the weights $\theta = \omega \in \Omega$. As in the previous example, $\theta$ does not depend on the task $\tau$. We remark that this example and Example 1 are similar in the sense that $\Phi$ is gradient descent and $\Psi$ is the constant function returning neural network's weights in both examples, though the task $\tau$ is different. The unified framework sheds light on a similarity in various methods for various tasks, which enables us to apply a technique developed for one problem to another easily.

Lastly, let us consider Chen et al. (2020). In this work, meta-learning is used to generate a parameter of PDE-MgNet, a neural network representing the multigrid method, for solving parametrized PDEs. This also can be regarded as a special case of GBMS, where $\tau$ is solving a PDE, the solver $\Phi$ is the PDE-MgNet, whose iterative function is implemented by a neural network $\phi$ with weights $\theta$, and the meta-solver $\Psi$ is another neural network with weights $\omega$ to generate $\theta$ depending on task $\tau$. Note that the meta-solver $\Psi$ is trained with single step of the solver $\Phi$ but tested with multiple steps of $\Phi$. In other words, this work does not consider the solver $\Phi$ itself but instead its iterative function $\phi$ in the training. We will show the importance of this difference in section 3.1.

The above examples show the generality of our meta-solving formulation that organizes a variety of methods in the systematic way regardless of their type of algorithm. This overall approach allows us to easily design learning-based methods for new problem settings, and we will demonstrate it in the next section.

## 3 GBMS FOR ITERATIVE METHODS

In this section, we demonstrate how GBMS can be used to yield effective meta-solvers for new problem settings. In particular, we consider the problem of accelerating iterative methods by adaptively choosing initial conditions. Iterative methods are powerful tools to solve computational problems. For example, the Jacobi method and SOR method are used to solve PDEs (Saad (2003)). In iterative methods, a function $\phi_\tau$, which depends on the task $\tau$, is iteratively applied to the current approximate solution to update it closer to the true solution until it reaches a criterion, such as a certain error tolerance or number of iterations. These methods require an initial guess as the starting point of the iterations, and the performance of the method depends on this choice. Thus, adaptively choosing initial guesses for solvers is an effective way to speed up the computation process.

Here, we apply GBMS to derive a meta-solver that produces effective initial guesses for iterative solvers. The meta-solving problem here can be viewed as a generalized version of Example 2 beyond Jacobi solvers. The task $\tau$ is any computational problem which can be solved by iterative

methods. For example, $\tau$ is solving a PDE as described in Example 2. The task distribution $(\mathcal{T}, P)$ is defined according to each problem. The solver $\Phi : \mathcal{T} \times \Theta \rightarrow \mathcal{U}$ is an iterative method with an initial guess $\theta = u^{(0)} \in \Theta$, iterative function $\phi_\tau$, and the number of iterations $k$, so $\Phi(\tau; u^{(0)}) = \phi_\tau^k(u^{(0)}) = u^{(k)}$ and $\Theta = \mathcal{U}$. The meta-solver $\Psi : \mathcal{T} \times \Omega \rightarrow \Theta$ is a function parametrized by $\omega \in \Omega$, which takes $\tau \in \mathcal{T}$ as an input and generates an initial guess $\theta_\tau = u_\tau^{(0)} \in \Theta$ for the solver $\Phi$. We implement $\Psi$ by a neural network, so $\omega \in \Omega$ is its weights. Then, $\Psi$ is trained to minimize the expectation of $L_\tau(u^{(k)})$ by the gradient descent. Since both $\Phi$ and $\Psi$ are implemented in a deep learning framework, $\nabla_\omega L_\tau(u^{(k)})$ can be computed by back-propagation. The entire process of the algorithm is presented in Algorithm 2.

Hereafter, we consider solving a PDE as a task of the meta-solving problem, but the method is applicable to other tasks, such as root finding. Note that although Huang et al. (2020), Ajuria Illarramendi et al. (2020), and Özbay et al. (2021) propose to use initial guesses generated by neural networks, these initial guesses are independent of the solvers. On the other hand, our initial guesses are optimized for each solver. We will show its advantage in the following sections.

---

**Algorithm 2:** Gradient-based meta-solving algorithm for iterative solvers

---

**Require:** $(P, \mathcal{T})$: task space, $\Psi$: differentiable solver, $S$: stopping criterion, $k$: number of
      iterations of the iterative solver, $\alpha$: learning rate
**Result:** $\omega$

1  initialize $\omega$;
2  **while** $S$ *is not satisfied* **do**
3      Sample task $\tau \sim P$;
4      Form iterative function $\phi_\tau$ depending on $\tau$;
5      Generate initial guess using meta-solver: $u_\tau^{(0)}(\omega) \leftarrow \Psi(\tau; \omega)$;
6      **for** $i \leftarrow 1$ *to* $k$ **do**
7          Update approximate solution by iterative function: $u_\tau^{(i)}(\omega) \leftarrow \phi_\tau(u_\tau^{(i-1)}(\omega))$;
8      **end**
9      Update meta-solver parameter: $\omega \leftarrow \omega - \alpha \nabla_\omega L_\tau(u_\tau^{(k)}(\omega))$;
10 **end**

---

### 3.1 TOY PROBLEM: 1D POISSON EQUATIONS

To study the property of the proposed algorithm, we consider solving 1D Poisson equations as a toy problem. First, we show a theorem that guarantees the improvement of the proposed meta-solving approach for the Jacobi method and linear neural networks. Then, we demonstrate that the theorem numerically holds for another iterative method and practical nonlinear neural network.

#### 3.1.1 THEORETICAL ANALYSIS

Let us recall Example 2 and set the domain of interest $\mathcal{D} = (0, 1)$. Then, the target equation becomes the following 1D Poisson equation with Dirichlet boundary condition:

$$-\frac{d^2}{dx^2} u(x) = f(x), \quad x \in (0, 1)$$
$$u(0) = a, \quad u(1) = b. \tag{2}$$

To solve this equation numerically, it is discretized with finite difference scheme, and we rewrite it as the matrix equation $Au = f$, where the domain $[0, 1]$ is discretized into $N$ points, so $A \in \mathbb{R}^{N \times N}$ and $u, f \in \mathbb{R}^N$. Suppose that we solve the equation $Au = f$ with the Jacobi method under randomly sampled $f_\tau$. Then, the meta-solving problem for solving 1D Poisson equations is defined by the following. The task $\tau$ is to solve $Au = f_\tau$. The dataset is $D_\tau = \{f_\tau, u_\tau\}$, where $u_\tau$ is the solution of $Au = f_\tau$. The solution parameter space is $\mathcal{U}_\tau = \mathbb{R}^N$. The loss function is $L_\tau(\hat{u}) = \|u_\tau - \hat{u}\|^2$. The task distribution $(\mathcal{T}, P)$ is determined by the distribution of $f_\tau$, denoted by $P_f$. It is assumed to be centered and normalized, i.e. the mean of $f_\tau$ is 0 and the covariance is the identity matrix. The solver $\Phi_k : \mathcal{T} \times \Theta \rightarrow \mathcal{U}$ is the Jacobi method with $k$ iterations starting at an initial guess

$u^{(0)} = \theta \in \Theta$. Its iterative function is $\phi_\tau(u) = Mu + \frac{1}{2}f_\tau$, where $M = I - \frac{1}{2}A$. To summarize, $\Phi_k(\tau; u^{(0)}) = \phi_\tau^k(u^{(0)}) = \hat{u}$. Note that $\Phi_0$ is the identity map. The meta-solver $\Psi : \mathcal{T} \times \Omega \to \Theta$ is a linear neural network with weights $\omega \in \Omega$. It can be represented as matrix multiplication for some matrix $W$, so $\Psi(\tau; \omega) = Wf_\tau = u^{(0)}$. Here, we assume $\mathrm{rank}(W) < N$ to avoid the trivial solution $W = A^{-1}$, i.e. the meta-solver does not have the capacity to produce an exact solution for each task. Then, the meta-solving problem becomes

$$\min_{\omega \in \Omega} \mathbb{E}_{\tau \sim P} [L_\tau(\Phi(\tau; \Psi(\tau; \omega))] = \min_W \mathbb{E}_{f_\tau \sim P_f} \left\| u_\tau - \phi_\tau^k(Wf_\tau) \right\|^2. \tag{3}$$

As for this meta-solving problem, the following theorem holds. The proof is in Appendix B.

**Theorem 1** (Guarantee of improvement by meta-solving). *For any $k \geq 0$, (3) has the unique minimizer $W_k$. Furthermore, if $k_1 < k_2$, then for all $k \geq k_2$,*

$$\mathbb{E}_{f_\tau \sim P_f} \left\| u_\tau - \phi_\tau^k(W_{k_1} f_\tau) \right\|^2 \geq \mathbb{E}_{f_\tau \sim P_f} \left\| u_\tau - \phi_\tau^k(W_{k_2} f_\tau) \right\|^2, \tag{4}$$

*where the equality holds if and only if $W_{k_1} = W_{k_2}$.*

Theorem 1 guarantees improvement of meta-solving for the considered setting. For example, suppose $k_1 = 0$ (regular supervised learning, i.e. directly predicting the solution without considering the solver) and $k_2 = 5$. Then, the theorem implies that the meta-solver trained with 5 Jacobi iterations is expected to give an better initial guess than the meta-solver obtained by regular supervised learning for any number of Jacobi iterations larger than 5. More generally, Theorem 1 shows that meta-learning with a higher number of Jacobi iterations in the inner solver improves the performance, provided one carries out at least a greater number of Jacobi iterations during inference. This shows the advantage of the meta-solving approach over regular supervised learning.

### 3.1.2 NUMERICAL EXAMPLES FOR MORE GENERAL CASES

The key insight from the previous theoretical analysis is that meta-solving leverages the properties of the solver and adapts the selection of initial conditions to it. Here, we show using numerical examples that this is the case for more complex scenarios, involving different task distributions, different iterative solvers, and a practical nonlinear neural network.

The meta-solving problem in this section is defined by the following. The task $\tau$ is the same as the previous section 3.1.1. Let the number of discretization points $N$ be 512. The task distribution $(\mathcal{T}, P)$ is determined by the distribution of $u$. We consider two distributions $P_s$ and $P_h$, where the solution $u$ consists of sine functions and hyperbolic tangent functions respectively. Their details are listed in Appendix C.1. For each distribution, we prepare 30,000 tasks for training, 10,000 for validation, and 10,000 for test. To solve the tasks, we use the Jacobi method and the Red-Black ordering SOR method (Saad (2003)) with $k$ iterations starting at an initial guess $u^{(0)} = \theta \in \Theta$, denoted by $\Phi_{\mathrm{Jac},k}$ and $\Phi_{\mathrm{SOR},k}$ respectively. Note that they are implemented by convolutional layers. We consider two meta-solvers. One is a heuristic initial guess generator $\Psi_{\mathrm{BL}}$ that takes $\tau$ as an input and gives the heuristic initial guess, which is the linear interpolation of the boundary condition. $\Psi_{\mathrm{BL}}$ does not have a parameter and is used as a baseline. The other is a variant of 1D U-Net (Ronneberger et al. (2015)) $\Psi_{\mathrm{NN}}$ with weights $\omega \in \Omega$, which takes $f_\tau$ and the heuristic initial guess $\Psi_{\mathrm{BL}}(\tau)$ as inputs and generates an initial guess $\theta_\tau \in \Theta$ for the solver $\Phi$.

The meta-solver $\Psi_{\mathrm{NN}}$ is trained with solvers $\Phi_{\mathrm{Jac},k}, \Phi_{\mathrm{SOR},k}$ with $k = 0, 4, 16, 64$ by using the Algorithm 2. For each setting, $\Psi_{\mathrm{NN}}$ is trained six times with different random seeds. The details of the architecture and hyper-parameters of $\Psi_{\mathrm{NN}}$ are found in Appendix D.1. The trained meta-solver $\Psi_{\mathrm{NN}}$ and baseline $\Psi_{\mathrm{BL}}$ are tested with solvers $\Phi_{\mathrm{Jac},k}, \Phi_{\mathrm{SOR},k}$ with $k = 0, 4, 16, 64$. The performances of the meta-solvers are measured by the MSE on the test set and presented in Table 1. Figure 1 shows the comparison of the initial guesses $u^{(0)}$ obtained by $\Psi_{\mathrm{NN}}$ with $\Phi_{\mathrm{Jac},0}$ and $\Phi_{\mathrm{Jac},64}$. Figure 2 is the convergence plot of $\Psi_{\mathrm{NN}}$ trained with $\Phi_{\mathrm{SOR},k}$ on $P_s$.

The results show that the meta-solver $\Psi_{\mathrm{NN}}$ is optimized for each corresponding solver $\Phi$. In Table 1, the best performance for a given test solver is achieved at the diagonal where the training and test solvers match. We can explicitly investigate this adaptive phenomena by visualizing the meta-solver outputs for a representative problem instance. Figure 1 shows that the meta-solver trained with more iterations tends to ignore high frequencies and focus more on low frequencies. This is because the

Jacobi method converges fast in high frequencies but slow in low frequencies (Saad (2003)), and the trained meta-solver compensates the weakness of the solver. These results show that the meta-solver actively adapts to the nature of the inner-loop solver.

Furthermore, we can observe that Theorem 1 holds numerically for the results. Figure 2 illustrates that a meta-solver trained with more iterations is always better than those trained with fewer iterations, provided the number of iterations during testing is large. In addition, at 10,000 iterations, the error of $\Psi_{\mathrm{NN}}$ trained with $\Phi_{\mathrm{SOR},64}$ (MSE of 0.0054 and relative error of 0.013) remains approximately 8% better than $\Psi_{\mathrm{NN}}$ trained with $\Phi_{\mathrm{SOR},0}$, which shows that the GBMS keeps its advantage for a large number of iterations. These results support that Theorem 1 holds for various practical settings with significant improvement.

Table 1: Average MSE of GBMS for solving Poisson equations. Boldface indicates best performance for each column. Standard deviations are presented in Table 3 in Appendix E.

(a) MSE on $P_s$

| $\Psi$ | Trained with | Tested with $\Phi_{\mathrm{Jac},0}$ | $\Phi_{\mathrm{Jac},4}$ | $\Phi_{\mathrm{Jac},16}$ | $\Phi_{\mathrm{Jac},64}$ | $\Phi_{\mathrm{SOR},4}$ | $\Phi_{\mathrm{SOR},16}$ | $\Phi_{\mathrm{SOR},64}$ |
|---|---|---|---|---|---|---|---|---|
| $\Psi_{\mathrm{NN}}$ | $\Phi_{\mathrm{Jac},0} = \Phi_{\mathrm{SOR},0}$ | **0.222** | 0.220 | 0.219 | 0.214 | 0.218 | 0.212 | 0.193 |
| | $\Phi_{\mathrm{Jac},4}$ | 0.277 | **0.212** | 0.210 | 0.205 | 0.210 | 0.203 | 0.186 |
| | $\Phi_{\mathrm{Jac},16}$ | 1.169 | 0.231 | **0.209** | 0.204 | 0.221 | 0.203 | 0.185 |
| | $\Phi_{\mathrm{Jac},64}$ | 17.684 | 3.066 | 0.279 | **0.198** | 0.299 | 0.198 | 0.179 |
| | $\Phi_{\mathrm{SOR},4}$ | 1.140 | 0.982 | 0.753 | 0.496 | **0.201** | 0.195 | 0.179 |
| | $\Phi_{\mathrm{SOR},16}$ | 25.786 | 6.145 | 1.134 | 0.554 | 0.392 | **0.185** | 0.170 |
| | $\Phi_{\mathrm{SOR},64}$ | 22.451 | 8.306 | 2.459 | 0.744 | 2.536 | 0.224 | **0.169** |
| $\Psi_{\mathrm{BL}}$ | - | 17.033 | 12.184 | 9.482 | 7.926 | 9.083 | 7.611 | 6.606 |

(b) MSE on $P_h$

| $\Psi$ | Trained with | Tested with $\Phi_{\mathrm{Jac},0}$ | $\Phi_{\mathrm{Jac},4}$ | $\Phi_{\mathrm{Jac},16}$ | $\Phi_{\mathrm{Jac},64}$ | $\Phi_{\mathrm{SOR},4}$ | $\Phi_{\mathrm{SOR},16}$ | $\Phi_{\mathrm{SOR},64}$ |
|---|---|---|---|---|---|---|---|---|
| $\Psi_{\mathrm{NN}}$ | $\Phi_{\mathrm{Jac},0} = \Phi_{\mathrm{SOR},0}$ | **1.049** | 1.041 | 1.030 | 1.005 | 1.028 | 0.994 | 0.922 |
| | $\Phi_{\mathrm{Jac},4}$ | 1.059 | **1.023** | **1.006** | 0.976 | 1.002 | 0.964 | 0.888 |
| | $\Phi_{\mathrm{Jac},16}$ | 1.202 | 1.049 | **1.006** | 0.971 | 1.000 | 0.958 | 0.885 |
| | $\Phi_{\mathrm{Jac},64}$ | 2.017 | 1.572 | 1.142 | **0.940** | 1.058 | 0.920 | 0.839 |
| | $\Phi_{\mathrm{SOR},4}$ | 1.632 | 1.479 | 1.383 | 1.333 | **0.984** | 0.938 | 0.862 |
| | $\Phi_{\mathrm{SOR},16}$ | 3.305 | 2.597 | 1.764 | 1.285 | 1.305 | **0.928** | 0.845 |
| | $\Phi_{\mathrm{SOR},64}$ | 7.683 | 6.009 | 3.580 | 1.417 | 3.029 | 1.116 | **0.756** |
| $\Psi_{\mathrm{BL}}$ | - | 10.761 | 10.744 | 10.695 | 10.539 | 10.681 | 10.463 | 9.908 |

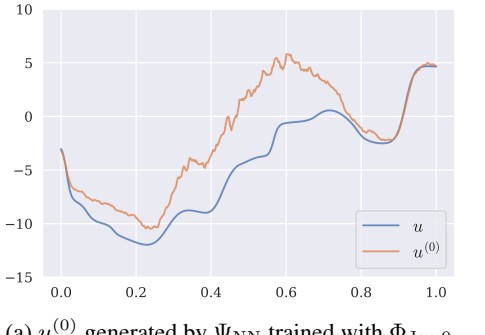

(a) $u^{(0)}$ generated by $\Psi_{\mathrm{NN}}$ trained with $\Phi_{\mathrm{Jac},0}$

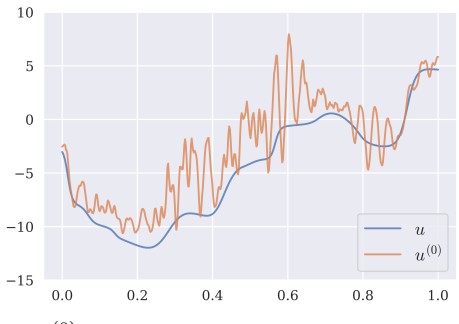

(b) $u^{(0)}$ generated by $\Psi_{\mathrm{NN}}$ trained with $\Phi_{\mathrm{Jac},64}$

Figure 1: Comparison of initial guesses $u^{(0)}$ generated by $\Psi_{\mathrm{NN}}$

## 3.2 APPLICATION: INCOMPRESSIBLE FLOW SIMULATIONS

In this section, we introduce an application of GBMS for iterative solvers. GBMS is developed for repeatedly solving task $\tau$ sampled from a given task distribution $P$. One of the typical situations where GBMS is effective is that similar differential equations are repeatedly solved as a step of solving a time-dependent problem. For example, in many methods of incompressible flow simula-

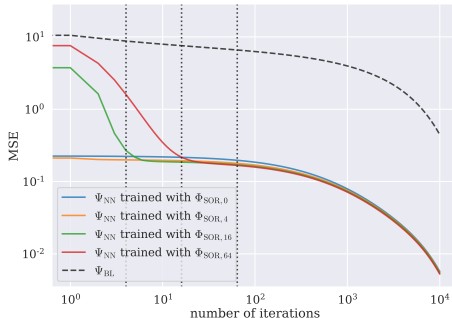

Figure 2: Convergence plot of $\Psi$ trained with $\Phi_{\mathrm{SOR},k}$ on $P_s$. Vertical dotted lines are $k = 4, 16, 64$.

tions, Poisson equations are solved to compute the pressure field at every time step, and this process occupies a large part of the computation time for fluid simulations (Ajuria Illarramendi et al. (2020)).

**Problem definition.** Let us now describe the incompressible flow simulation setup. We consider 2D channel flow with a fluctuating inflow. The velocity field $v$ and pressure field $p$ follow the Navier-Stokes equations in the form:

$$\frac{\partial v}{\partial t} + (v \cdot \nabla)v = -\frac{1}{\rho}\nabla p + \nu \Delta v \tag{5}$$

$$\Delta p = -\rho \nabla \cdot ((v \cdot \nabla)v) \tag{6}$$

where $\rho$ is the density, and $\nu$ is the viscosity. The equation (6) is called the pressure Poisson equation. Let $\rho = 1$, $\nu = 0.01$. Let the domain of interest $\mathcal{D}$ be $(0,1) \times (0,1)$ and the time $t \in [0,1]$. Let $y = 0$ and $y = 1$ be the non-slip wall and fluctuating inflow $v_{\mathrm{in}}$ defined on $x = 0$. By using the finite difference scheme, we discretize the equations into 128 by 128 spacial grids and 1,000 time steps. Then, $v$ and $p$ are computed alternately by the discretized equations. Details of the numerical scheme for solving Navier-Stokes equations follow Barba & Forsyth (2018). GBMS for iterative method is used for solving the pressure Poisson equation (6).

Let us define the meta-solving problem for the above setting. The task $\tau$ is to solve a pressure Poisson equation. The dataset $D_\tau$ is $\{f_\tau, p_\tau, f_{\tau-1}, p_{\tau-1}, f_{\tau-2}, p_{\tau-2}\}$, where $f_\tau$ is the right hand side and $p_\tau$ is the solution of the pressure Poisson equation. $f_{\tau-i}, p_{\tau-i}$ are those at $i$th previous time step. Although $p_\tau$ is determined by $f_\tau$ theoretically, additional features of previous timesteps can provide useful information to determine a good guess for $p_\tau$. The solution parameter space $\mathcal{U}_\tau$ is $\mathbb{R}^{128 \times 128}$. The loss function is the relative $\ell^2$-error. The task distribution $(\mathcal{T}, P)$ is determined by the distribution of $v_{\mathrm{in}}$ and the computation process of the simulation. Let $P_{\mathrm{in}}$ be the distribution of $v_{\mathrm{in}}$, whose detail is found in Appendix C.2. We prepare 20 inflows for training, 10 for validation, and 10 for test, which are denoted by $\mathcal{V}_{\mathrm{in}}^{\mathrm{train}}, \mathcal{V}_{\mathrm{in}}^{\mathrm{val}}$ and $\mathcal{V}_{\mathrm{in}}^{\mathrm{test}}$ respectively. Then, we generate datasets under each inflow by solving the Navier-Stokes equations with the SOR method without GBMS, where the relative error tolerance is $10^{-9}$. The solver $\Phi_{\mathrm{SOR},k}$ is the SOR method with $k$ iterations starting at an initial guess $\theta \in \Theta$. We consider two meta-solvers. One is a baseline $\Psi_{\mathrm{BL}}$ that takes $\tau$ as an input and gives the heuristic initial guess, which is the solution of the previous step Poisson equation $p_{\tau-1}$. This choice is based on the domain knowledge of fluid simulations and standard in the literature (Ferziger & Perić (2002)). The meta-solver $\Psi_{\mathrm{NN}}$ is a variant of 2D U-Net with weights $\omega \in \Omega$, which takes $\mathcal{D}_\tau \setminus \{p_\tau\}$ as an input and generates initial guess $\theta_\tau \in \Theta$ for the solver $\Phi_{\mathrm{SOR},k}$. Note that $\Psi_{\mathrm{NN}}$ trained with $\Phi_{\mathrm{SOR},0}$ (i.e. solver-independent initial guess) is similar to the method in Ajuria Illarramendi et al. (2020) and is considered as a data-driven baseline in this paper. We also note that the choice of $\Psi_{\mathrm{NN}}$ is arbitrary, and any other problem-specific neural network architectures can be used as $\Psi_{\mathrm{NN}}$.

**Training.** We train the meta-solver $\Psi_{\mathrm{NN}}$ with the SOR solver $\Phi_{\mathrm{SOR},k}$ with $k = 0, 4, 16, 64$ by using the Algorithm 2. The details of the architecture and hyper-parameters of $\Psi_{\mathrm{NN}}$ are found in Appendix D.2. In addition, we use data augmentation during training to prevent overfitting to the high-accuracy training data. During the training, we augment data by proceeding 10 time steps with $\Phi_{\mathrm{SOR},k}$ and $\Psi_{\mathrm{NN}}$ being trained. Specifically, after computing the loss for $p_\tau$, we proceed one time step using $\Phi_{\mathrm{SOR},k}$ and $\Psi_{\mathrm{NN}}$, and compute the loss for next $p_{\tau+1}$ using the computed previous step information.

Then, it is repeated 10 times for each mini batch. Without the data augmentation, the inputs of $\Psi_{\text{NN}}$ are from the prepared dataset and are always accurate during training. However, during inference, the inputs of $\Psi_{\text{NN}}$ are the outputs of the simulation at previous time steps, which can be of a slightly different distribution than encountered during training (i.e. distribution shift) due to numerical errors and generalization gaps. This then causes further accuracy problems for the simulation at the next step, and this issue compounds itself in time. This eventually degrades the performance if the shift in distribution is not dealt with. The data augmentation is used to prevent it. We remark that this training process does not require the whole of the simulation to be differentiable. This is practically important because we can employ GBMS with established solvers by only replacing the Poisson solver and can utilize the other parts without any modification.

**Evaluation.** We incorporate the trained meta-solvers into simulations and evaluate them by the relative $\ell^2$-error of the velocity field on test inflows under a fixed number of the SOR iterations. More specifically, the performance metric is $\frac{1}{|\mathcal{V}_{\text{in}}^{\text{test}}|} \sum_{v_{\text{in}} \in \mathcal{V}_{\text{in}}^{\text{test}}} \frac{1}{900} \sum_{i=101}^{1000} \frac{\|v_i - \hat{v}_i\|}{\|\hat{v}_i\|}$, where $v_i$ is the ground truth of $i$th step velocity under the inflow $v_{\text{in}}$, which is obtained by using $\Psi_{\text{BL}}$ and $\Phi_{\text{SOR},k}$ with a large enough $k$ to achieve the relative error tolerance of $10^{-9}$, and $\hat{v}_i$ is the $i$th step velocity obtained by using the trained $\Psi_{\text{NN}}$ and $\Phi_{\text{SOR},k}$ with a fixed $k$. To stabilize the simulation, the first 100 steps are excluded, and the simulation with the trained $\Psi_{\text{NN}}$ starts with 101 steps.

The results presented in Table 2 demonstrate the advantage of GBMS. Although all $\Psi_{\text{NN}}$ diverge for $\Phi_{\text{SOR},0}$ because of error accumulation, for the other solvers, the best performance with significant improvement is achieved at the diagonal where the training and test solvers match. For example, for the test solver $\Phi_{\text{SOR},4}$, the accuracy of $\Psi_{\text{NN}}$ trained with $\Phi_{\text{SOR},4}$ is approximately 50 times better than regular supervised learning ($\Psi_{\text{NN}}$ trained with $\Phi_{\text{SOR},0}$) and 130 times better than the classical baseline $\Psi_{\text{BL}}$. Furthermore, GBMS keeps its advantage for a larger number of iterations than the number which it is trained with. For example, for $\Phi_{\text{SOR},64}$, the meta-solver trained with $\Phi_{\text{SOR},64}$ has the best performance, followed by one trained with $\Phi_{\text{SOR},16}$, $\Phi_{\text{SOR},4}$, and $\Phi_{\text{SOR},0}$ in that order. As for computation time, $\Psi_{\text{NN}}$ trained with $\Phi_{\text{SOR},4}$ and tested with $\Phi_{\text{SOR},4}$ (GBMS approach) takes 11 sec to simulate the flow for one second, while $\Psi_{\text{BL}}$ tested with $\Phi_{\text{SOR},128}$ takes 105 sec to achieve similar accuracy to the GBMS approach. Also, $\Psi_{\text{NN}}$ trained with $\Phi_{\text{SOR},0}$ and tested with $\Phi_{\text{SOR},64}$ takes 73 sec to achieve the similar accuracy. Since it takes approximately 7 hours to train $\Psi_{\text{NN}}$ with a GeForce RTX 3090, our approach is effective if we simulate the flow for more than 268 seconds, which can be easily satisfied in practical settings. To summarize, GBMS performs best for a fixed number of iterations and can achieve high accuracy within a smaller number of iterations and shorter computation time than the classical baseline and regular supervised learning.

Table 2: Relative $\ell^2$-error of GBMS on incompressible flow simulations.

| $\Psi$ | Trained with | Tested with | | | |
|---|---|---|---|---|---|
| | | $\Phi_{\text{SOR},0}$ | $\Phi_{\text{SOR},4}$ | $\Phi_{\text{SOR},16}$ | $\Phi_{\text{SOR},64}$ |
| $\Psi_{\text{NN}}$ | $\Phi_{\text{SOR},0}$ | NaN | 0.054861 | 0.008543 | 0.001284 |
| | $\Phi_{\text{SOR},4}$ | NaN | **0.001171** | 0.001625 | 0.000864 |
| | $\Phi_{\text{SOR},16}$ | NaN | 0.017513 | **0.000829** | 0.000723 |
| | $\Phi_{\text{SOR},64}$ | NaN | NaN | 0.005567 | **0.000312** |
| $\Psi_{\text{BL}}$ | - | 1.988177 | 0.152626 | 0.014112 | 0.002941 |

## 4 CONCLUSION

In this paper, we proposed a formulation of meta-solving and a general gradient-based algorithm (GBMS) to solve this class of problems. In the proposed framework, many related works that used neural networks for solving differential equations were organized in a unified way and regarded as variants of GBMS. Thus, the GBMS approach offers a general design pattern to develop solution algorithms that blends machine learning and scientific computing. As a concrete illustration, we applied GBMS to iterative methods for solving differential equations and showed its advantage over both classical numerical methods and regular supervised learning. In particular, the proposed method was tested in the incompressible flow simulation and the accuracy was improved by approximately 50 times compared to regular supervised learning and 130 times compared to a classical baseline for a fixed number of iterative solver iterations. We will study applications of GBMS to other types of problems such as nonlinear equations in future work.

## 5 REPRODUCIBILITY STATEMENT

To ensure the reproducibility, we provide the detailed proof of Theorem 1 in Appendix B. As for the experiments, we provide the details of dataset generation in Appendix C and the details of network architecture and training hyper-parameters in Appendix D. In addition, we conducted experiments with different random seeds to obtain robust results. Finally, we will make our source code public at a later date. Source code with limited documentation is available upon request.

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

# A  DETAILS OF EXAMPLES

**Example 3** (Feliu-Fabà et al. (2020)). Feliu-Fabà et al. (2020) propose the neural network architecture with meta-learning approach that solves the equations in the form $\mathcal{L}_\eta u(x) = f(x)$ with appropriate boundary conditions, where $\mathcal{L}_\eta$ is a partial differential or integral operator parametrized by a parameter function $\eta(x)$. This work can be described as follows:

- Task $\tau$: The task $\tau$ is to solve a $\mathcal{L}_\eta u(x) = f(x)$ for $\eta = \eta_\tau$:

  - Dataset $D_\tau$: The dataset $D_\tau$ is $D_\tau = \{\eta_\tau, f_\tau, u_\tau\}$, where $\eta_\tau, f_\tau, u_\tau \in \mathbb{R}^N$ are the [], right hand side, and solution respectively. The reference solution $u_\tau$ is obtained by [].
  - Solution parameter space $\mathcal{U}_\tau$: The solution parameter space $\mathcal{U}_\tau$ is a subset of $\mathbb{R}^N$ for $N \in \mathbb{N}$.
  - Loss function $L_\tau$: The loss function $L_\tau : \mathcal{U} \to \mathbb{R}_{\geq 0}$ is the mean squared error with the reference solution, i.e. $L_\tau(\hat{u}) = \|u_\tau - \hat{u}\|^2$.

- Task space $(\mathcal{T}, P)$: The task distribution $(\mathcal{T}, P)$ is determined by the distribution of $\eta_\tau$ and $f_\tau$.

- Solver $\Phi$: The solver $\Phi : \mathcal{T} \times \Theta \to \mathcal{U}_\tau$ is implemented by a neural network imitating the wavelet transform, which is composed by three modules with weights $\theta = (\theta_1, \theta_2, \theta_3)$. In detail, the three modules, $\phi_1(\cdot; \theta_1)$, $\phi_2(\cdot; \theta_2)$, and $\phi_3(\cdot; \theta_3)$, represent forward wavelet transform, mapping $\eta$ to coefficients matrix of the wavelet transform, and inverse wavelet transform respectively. Then, $\Phi$ is represented by $\Phi(\tau; \theta) = \phi_3((\phi_2(\eta_\tau; \theta_2)\phi_1(f_\tau; \theta_1)); \theta_3) = \hat{u}$.

- Meta-solver $\Psi$: The meta-solver $\Psi : \mathcal{T} \times \Omega \to \Theta$ is the constant function that returns its parameter $\omega$, so $\Psi(\tau; \omega) = \omega = \theta$ and $\Omega = \Theta$. Note that $\theta$ does not depend on $\tau$ in this example.

Then, the weights $\omega = \theta$ is optimized by a gradient descent algorithm as described in Algorithm 1.

**Example 4** (Chen et al. (2020)). The target equation in Chen et al. (2020) is a linear systems of equations $A_\eta u = f$ obtained by discretizing parameterized steady-state PDEs, where $u, f \in \mathbb{R}^N$ and $A_\eta \in \mathbb{R}^{N \times N}$ is determined by $\eta$, a parameter of the original equation. This work can be described as follows:

- Task $\tau$: The task $\tau$ is to solve a linear system $A_\eta u = f$ for $\eta = \eta_\tau$:

  - Dataset $D_\tau$: The dataset $D_\tau$ is $\{\eta_\tau, f_\tau\}$.
  - Solution parameter space $\mathcal{U}_\tau$: The solution parameter space $\mathcal{U}_\tau$ is $\mathbb{R}^N$.
  - Loss function $L_\tau$: The loss function $L_\tau : \mathcal{U} \to \mathbb{R}_{\geq 0}$ is an unsupervised loss based on the residual of the equation, $L_\tau(\hat{u}) = \|f_\tau - A_{\eta_\tau}\hat{u}\|^2 / \|f_\tau\|^2$.

- Task space $(\mathcal{T}, P)$: The task distribution $(\mathcal{T}, P)$ is determined by the distribution of $\eta_\tau$ and $f_\tau$.

- Solver $\Phi$: The solver $\Phi : \mathcal{T} \times \Theta \to \mathcal{U}$ is iterations of a function $\phi_\tau(\cdot; \theta) : \mathcal{U} \to \mathcal{U}$ that represents an update step of the multigrid method. $\phi_\tau$ is implemented using a convolutional neural network and its parameter $\theta$ is the weights corresponding to the smoother of the multigrid method. Note that weights of $\phi_\tau$ other than $\theta$ are naturally determined by $\eta_\tau$ and the discretization scheme. In addition, $\phi_\tau$ takes $f_\tau$ as part of its input at every step, but we write these dependencies as $\phi_\tau$ for simplicity. To summarize, $\Phi(\tau; \theta) = \phi_\tau^k(u^{(0)}; \theta) = \hat{u}$, where $k$ is the number of iterations of the multigrid method and $u^{(0)}$ is initial guess, which is **0** in the paper.

- Meta-solver $\Psi$: The meta-solver $\Psi : \mathcal{T} \times \Omega \to \Theta$ is implemented by a neural network with weights $\omega$, which takes $A_{\eta_\tau}$ as its input and returns weights $\theta_\tau$ that is used for the smoother inspired by the subspace correction method.

Then, $\omega$ is optimized by a gradient decent algorithm with the number of multigrid iteration $k = 1$ as described in Algorithm 1.

**Example 5** (Psaros et al. (2021)). In Psaros et al. (2021), meta-learning is used for learning a loss function of the physics-informed neural network, shortly PINN (Raissi et al. (2019)). The target equations are the following:

$$\mathcal{F}_\lambda[u](t,x) = 0, (t,x) \in [0,T] \times \mathcal{D} \tag{a}$$

$$\mathcal{B}_\lambda[u](t,x) = 0, (t,x) \in [0,T] \times \partial\mathcal{D} \tag{b}$$

$$u(0,x) = u_{0,\lambda}(x), x \in \mathcal{D}, \tag{c}$$

where $\mathcal{D} \subset \mathbb{R}^M$ is a bounded domain, $u : [0,T] \times \mathcal{D} \to \mathbb{R}^N$ is the solution, $\mathcal{F}_\lambda$ is a nonlinear operator containing differential operators, $\mathcal{B}_\lambda$ is a operator representing the boundary condition, $u_{0,\lambda} : \mathcal{D} \to \mathbb{R}^N$ represents the initial condition, and $\lambda$ is a parameter of the equations.

- Task $\tau$: The task $\tau$ is to solve a differential equation by PINN:

    - Dataset $D_\tau$: The dataset $D_\tau$ is the set of points $(t,x) \in [0,T] \times \mathcal{D}$ and the values of $u$ at the points if applicable. In detail, $D_\tau = D_{f,\tau} \cup D_{b,\tau} \cup D_{u_0,\tau} \cup D_{u,\tau}$, where $D_{f,\tau}, D_{b,\tau}$, and $D_{u_0,\tau}$ are sets of points corresponding to the equation (a), (b), and (c) respectively. $D_{u,\tau}$ is the set of points $(t,x)$ and observed values $u(t,x)$ at the points. In addition, each dataset $D_{\cdot,\tau}$ is divided into training set $D_{\cdot,\tau}^{\text{train}}$ and validation set $D_{\cdot,\tau}^{\text{val}}$.

    - Solution parameter space $\mathcal{U}_\tau$: The solution parameter space $\mathcal{U}_\tau$ is the weights space of PINN.

    - Loss function $L_\tau$: The loss function $L_\tau : \mathcal{U} \to \mathbb{R}_{\geq 0}$ is based on the evaluations at the points in $D_\tau^{\text{val}}$. In detail,

    $$L_\tau(\hat{u}) = L_\tau^{\text{val}}(\hat{u}) = L_{f,\tau}^{\text{val}}(\hat{u}) + L_{b,\tau}^{\text{val}}(\hat{u}) + L_{u_0,\tau}^{\text{val}}(\hat{u}),$$

    where

    $$L_{f,\tau}^{\text{val}} = \frac{w_f}{|D_{f,\tau}|} \sum_{(t,x) \in D_{f,\tau}} \ell\left(\mathcal{F}_\lambda[\hat{u}](t,x), \mathbf{0}\right)$$

    $$L_{b,\tau}^{\text{val}} = \frac{w_b}{|D_{b,\tau}|} \sum_{(t,x) \in D_{b,\tau}} \ell\left(\mathcal{B}_\lambda[\hat{u}](t,x), \mathbf{0}\right)$$

    $$L_{u_0,\tau}^{\text{val}} = \frac{w_{u_0}}{|D_{u_0,\tau}|} \sum_{(t,x) \in D_{u_0,\tau}} \ell\left(\hat{u}(0,x), u_{0,\lambda}(x)\right),$$

    and $\ell : \mathbb{R}^N \times \mathbb{R}^N \to \mathbb{R}_{\geq 0}$ is a function. In the paper, the mean squared error is used as $\ell$.

- Task space $(\mathcal{T}, P)$: The task distribution $(\mathcal{T}, P)$ is determined by the distribution of $\lambda$.

- Solver $\Phi$: The solver $\Phi : \mathcal{T} \times \Theta \to \mathcal{U}_\tau$ is the gradient decent for training the PINN. The parameter $\theta \in \Theta$ controls the objective of the gradient decent, $L_\tau^{\text{train}}(\hat{u};\theta) = L_{f,\tau}^{\text{train}}(\hat{u};\theta) + L_{b,\tau}^{\text{train}}(\hat{u};\theta) + L_{u_0,\tau}^{\text{train}}(\hat{u};\theta) + L_{u,\tau}^{\text{train}}(\hat{u};\theta)$, where the difference from $L_\tau^{\text{val}}$ is that parametrized loss $\ell_\theta$ is used in $L_\tau^{\text{train}}$ instead of the MSE in $L_\tau^{\text{val}}$. Note that the loss weights $w_f, w_b, w_{u_0}, w_u$ in $L_\tau^{\text{train}}$ are also considered as part of the parameter $\theta$. In the paper, two designs of $\ell_\theta$ are studied. One is using a neural network, and the other is using a learned adaptive loss function. In the former design, $\theta$ is the weights of the neural network, and in the latter design, $\theta$ is the parameter in the adaptive loss function.

- Meta-solver $\Psi$: The meta-solver $\Psi : \mathcal{T} \times \Omega \to \Theta$ is the constant function that returns its parameter $\omega$, so $\Psi(\tau;\omega) = \omega = \theta$ and $\Omega = \Theta$. Note that $\theta$ does not depend on $\tau$ in this example.

Then, the parameter $\omega = \theta$ is optimized by a gradient decent algorithm as described in Algorithm 1.

# B  PROOF OF THEOREM 1

*Proof of Theorem 1.* Equation (3) can be represented as follows:

$$\min_{W} \mathbb{E}_{f_\tau \sim P_f} \left\| u_\tau - \phi_\tau^k(W f_\tau) \right\|^2 = \min_{W} \mathbb{E}_{f_\tau \sim P_f} \left\| M^k(u_\tau - W f_\tau) \right\|^2 \tag{7}$$

$$= \min_{W} \mathbb{E}_{f_\tau \sim P_f} \left\| M^k(A^{-1} f_\tau - W f_\tau) \right\|^2. \tag{8}$$

Since the mean of $f_\tau$ is 0 and the covariance is the identity matrix, $W$ is the minimizer of (8) if and only if $W$ is the minimizer of the following:

$$\min_{W} \left\| M^k(A^{-1} - W) \right\|_F^2. \tag{9}$$

Let $\lambda_i$ and $v_i$ ($i = 1, 2, \ldots, N$) be eigenvalues and eigenvectors of $M$. Note that $\lambda_i = \cos \frac{i\pi}{N+1}$ and $1 > \lambda_1 > \lambda_2 > \cdots > \lambda_N > -1$. Since $A = 2I - 2M$, eigenvalues and eigenvectors of $A$ are $2 - 2\lambda_i$ and $v_i$. By the eigenvalue decomposition, $M$ and $A$ can be written as

$$M = V \Lambda V^T \tag{10}$$

$$A = V(2I - 2\Lambda)V^T, \tag{11}$$

where $\Lambda = \text{diag}(\lambda_i)$ and $V = (v_1, v_2, \ldots, v_N)$. By the decompositions,

$$\left\| M^k(A^{-1} - W) \right\|_F^2 = \left\| V \Lambda^k V^T (V(2I - 2\Lambda)^{-1} V^T - W) \right\|_F^2 \tag{12}$$

$$= \left\| V \Lambda^k (2I - 2\Lambda)^{-1} V^T - V \Lambda^k V^T W \right\|_F^2 \tag{13}$$

Thus, the minimizer $W_k$ is

$$W_k = \sum_{j=1}^{r} (2 - 2\lambda_{i(j,k)})^{-1} v_{i(j,k)} v_{i(j,k)}^T, \tag{14}$$

where $r = \text{rank}(W_k) < N$, and $i(j, k)$ is the index $i$ where $\left| \frac{\lambda_i^k}{2 - 2\lambda_i} \right|$ takes the top $j$th value in $i \in \{1, 2, \ldots, N\}$. Then, the following lemma holds.

**Lemma 2.** *If $k_1 < k_2$, then for all $k \geq k_2$*

$$\left\| M^k(A^{-1} - W_{k_1}) \right\|_F^2 \geq \left\| M^k(A^{-1} - W_{k_2}) \right\|_F^2, \tag{15}$$

*where the equality holds if and only if $W_{k_1} = W_{k_2}$.*

Since (8) and (9) are equivalent, if Lemma 2 holds, then Theorem 1 holds.

*Proof of Lemma 2.* Let

$$I_k := \{1, 2, \ldots, N\} \setminus \{i(j, k) : j = 1, 2, \ldots, r\}. \tag{16}$$

$I_k$ is the set of $N - r$ indices corresponding to the least $N - r$ values of $\left| \frac{\lambda_i^k}{2 - 2\lambda_i} \right|$. We have

$$\left\| M^k(A^{-1} - W_{k_1}) \right\|_F^2 - \left\| M^k(A^{-1} - W_{k_2}) \right\|_F^2 \tag{17}$$

$$= \sum_{i \in I_{k_1}} \left( \frac{\lambda_i^k}{2 - 2\lambda_i} \right)^2 - \sum_{i \in I_{k_2}} \left( \frac{\lambda_i^k}{2 - 2\lambda_i} \right)^2 \tag{18}$$

$$= \sum_{i \in I_{k_1} \setminus I_{k_2}} \left( \frac{\lambda_i^k}{2 - 2\lambda_i} \right)^2 - \sum_{i \in I_{k_2} \setminus I_{k_1}} \left( \frac{\lambda_i^k}{2 - 2\lambda_i} \right)^2. \tag{19}$$

If $W_{k_1} = W_{k_2}$, then (19) equals 0. Assume $W_{k_1} \neq W_{k_2}$, so $I_{k_1} \neq I_{k_2}$. Note that $I_k = \{\min I_k, \min I_k + 1, \ldots, \min I_k + (N - r) - 1 = \max I_k\}$, and for any $i \in I_k$, we have $|\lambda_i| \leq |\lambda_{\max I_k}|$. In addition, for $k_1 < k_2$, we have $\max I_{k_1} \geq \max I_{k_2}$. Let

$$i_{k_1} := \underset{i \in I_{k_1} \setminus I_{k_2}}{\arg\min} \left| \frac{\lambda_i^k}{2 - 2\lambda_i} \right| \tag{20}$$

$$i_{k_2} := \underset{i \in I_{k_2} \setminus I_{k_1}}{\arg\max} \left| \frac{\lambda_i^k}{2 - 2\lambda_i} \right| \tag{21}$$

$$C := |I_{k_1} \setminus I_{k_2}| = |I_{k_2} \setminus I_{k_1}|. \tag{22}$$

Then, we have

$$(19) \geq C \left( \frac{\lambda_{i_{k_1}}^k}{2 - 2\lambda_{i_{k_1}}} \right)^2 - C \left( \frac{\lambda_{i_{k_2}}^k}{2 - 2\lambda_{i_{k_2}}} \right)^2 \tag{23}$$

$$= C \left( \left( \frac{\lambda_{i_{k_1}}^{k_2 + (k - k_2)}}{2 - 2\lambda_{i_{k_1}}} \right)^2 - \left( \frac{\lambda_{i_{k_2}}^{k_2 + (k - k_2)}}{2 - 2\lambda_{i_{k_2}}} \right)^2 \right) \tag{24}$$

$$> 0. \tag{25}$$

The last inequality (25) is because for any $i \in I_{k_2}$,

$$\left| \frac{\lambda_{i_{k_1}}^{k_2}}{2 - 2\lambda_{i_{k_1}}} \right| > \left| \frac{\lambda_i^{k_2}}{2 - 2\lambda_i} \right| \tag{26}$$

and

$$|\lambda_{i_{k_1}}| > |\lambda_{\max I_{k_2}}| \geq |\lambda_i|. \tag{27}$$

This completes the proof of Lemma 2. $\qquad\square$

Lemma 2 holds. Thus, Theorem 1 holds. $\qquad\square$

## C  DETAILS OF TASK DISTRIBUTIONS

### C.1  DISTRIBUTIONS OF 1D POISSON EQUATION

In $P_s$, $u$ is represented by

$$u(x) = \sum_{i=1}^{20} a_i \sin(b_i \pi (x - c_i)), \tag{28}$$

where

$$a_i \sim \mathcal{N}(0, 1) \tag{29}$$
$$b_i \sim \mathrm{Unif}(0, 128) \tag{30}$$
$$c_i \sim \mathrm{Unif}(0, 1). \tag{31}$$

In $P_h$, $u$ is represented by

$$u(x) = \sum_{i=1}^{20} a_i \tanh(b_i \pi (x - c_i)), \tag{32}$$

where

$$a_i \sim \mathcal{N}(0, 10) \tag{33}$$
$$b_i \sim \mathrm{Unif}(0, 30) \tag{34}$$
$$c_i \sim \mathrm{Unif}(0, 1). \tag{35}$$

## C.2 DISTRIBUTION OF INFLOW

In $P_{\text{in}}$, $v_{\text{in}} = (v_1, v_2)^T$ is represented by

$$v_1(y, t) = w(y, t) \cos z(t) \tag{36}$$
$$v_2(y, t) = w(y, t) \sin z(t), \tag{37}$$

where

$$w(y, t) = 0.5 + 0.5 \sin(ay - bt)\pi \sin y\pi \sin t\pi \tag{38}$$
$$z(t) = e \sin(ct - d)\pi \tag{39}$$

and

$$a \sim \text{Unif}(0, 10) \tag{40}$$
$$b \sim \text{Unif}(-5, 5) \tag{41}$$
$$c \sim \text{Unif}(0, 5) \tag{42}$$
$$d \sim \text{Unif}(0, 2) \tag{43}$$
$$e \sim \text{Unif}(0, \frac{\pi}{2}). \tag{44}$$

# D DETAILS OF NETWORK ARCHITECTURE AND HYPER-PARAMETERS

## D.1 DETAILS OF SECTION 3.1.2

In section 3.1.2, $\Psi_{\text{NN}}$ is a variant of 1D U-Net with a residual connection to leverage the heuristic initial guess $u_h = \Psi_{\text{BL}}(\tau)$, so $\Psi_{\text{NN}}(\tau) = \Psi_{\text{NN}}(f_\tau, u_h) = u_h + 1\text{DUNet}(f_\tau, u_h)$, where 1DUNet consists of four stages with halved resolutions. Each stage has two convolutional layers with kernel size 11 and the activation function $\tanh$. In the first stage, the number of channels is 8, and it is doubled as the resolution is halved. By utilizing the linearity of the Poisson equation, inputs are normalized by $\|f_\tau\|$ before feeding them into $\Psi_{\text{NN}}$, and the final output $\hat{u}$ is denormalized by $\|f_\tau\|$. The model is trained for 2000 epochs by Adam with the learning rate 0.0005 and the batch-size 512. The learning rate is decreased by $1/5$ times when the validation loss does not improve for 200 epochs, and the training is terminated when the validation loss does not improve for 250 epochs. The model with the best validation loss is used for evaluation.

## D.2 DETAILS OF SECTION 3.2

In section 3.2, $\Psi_{\text{NN}}$ is a variant of 2D U-Net. The differences from $\Psi_{\text{NN}}$ in section 3.1.2 are its dimension, kernel size 5, and the starting number of channels 16. The model is trained for 50 epochs by Adam with the learning rate 0.0005 and the batch-size 64, and the learning rate is decreased by $1/5$ at 30 epoch. Note that the meta-solver is trained for 10 time steps for each mini batch by the data augmentation. The model with the best validation loss is used for evaluation.

# E DETAILS OF EXPERIMENT RESULTS

Table 3: Standard deviation of MSE of GBMS for solving Poisson equations.

(a) Standard deviation of MSE on $P_s$

| $\Psi$ | Trained with | Tested with $\Phi_{\text{Jac},0}$ | $\Phi_{\text{Jac},4}$ | $\Phi_{\text{Jac},16}$ | $\Phi_{\text{Jac},64}$ | $\Phi_{\text{SOR},4}$ | $\Phi_{\text{SOR},16}$ | $\Phi_{\text{SOR},64}$ |
|---|---|---|---|---|---|---|---|---|
| $\Psi_{\text{NN}}$ | $\Phi_{\text{Jac},0} = \Phi_{\text{SOR},0}$ | 0.0052 | 0.0052 | 0.0052 | 0.0050 | 0.0051 | 0.0049 | 0.0043 |
| | $\Phi_{\text{Jac},4}$ | 0.0097 | 0.0101 | 0.0098 | 0.0093 | 0.0098 | 0.0091 | 0.0074 |
| | $\Phi_{\text{Jac},16}$ | 0.5548 | 0.0038 | 0.0036 | 0.0033 | 0.0095 | 0.0024 | 0.0020 |
| | $\Phi_{\text{Jac},64}$ | 4.8458 | 0.7034 | 0.0167 | 0.0037 | 0.0381 | 0.0040 | 0.0029 |
| | $\Phi_{\text{SOR},4}$ | 0.1211 | 0.1005 | 0.0739 | 0.0734 | 0.0085 | 0.0080 | 0.0067 |
| | $\Phi_{\text{SOR},16}$ | 10.4207 | 1.9210 | 0.2923 | 0.2396 | 0.0917 | 0.0072 | 0.0056 |
| | $\Phi_{\text{SOR},64}$ | 4.8430 | 0.9012 | 0.3301 | 0.2788 | 0.1937 | 0.0095 | 0.0078 |

(b) Standard deviation of MSE on $P_h$

| $\Psi$ | Trained with | Tested with $\Phi_{\text{Jac},0}$ | $\Phi_{\text{Jac},4}$ | $\Phi_{\text{Jac},16}$ | $\Phi_{\text{Jac},64}$ | $\Phi_{\text{SOR},4}$ | $\Phi_{\text{SOR},16}$ | $\Phi_{\text{SOR},64}$ |
|---|---|---|---|---|---|---|---|---|
| $\Psi_{\text{NN}}$ | $\Phi_{\text{Jac},0} = \Phi_{\text{SOR},0}$ | 0.0194 | 0.0192 | 0.0189 | 0.0184 | 0.0187 | 0.0181 | 0.0186 |
| | $\Phi_{\text{Jac},4}$ | 0.0273 | 0.0261 | 0.0258 | 0.0251 | 0.0257 | 0.0248 | 0.0223 |
| | $\Phi_{\text{Jac},16}$ | 0.0588 | 0.0212 | 0.0195 | 0.0208 | 0.0198 | 0.0213 | 0.0240 |
| | $\Phi_{\text{Jac},64}$ | 0.2278 | 0.1431 | 0.0513 | 0.0279 | 0.0369 | 0.0281 | 0.0288 |
| | $\Phi_{\text{SOR},4}$ | 0.6228 | 0.6871 | 0.7175 | 0.7154 | 0.0286 | 0.0308 | 0.0324 |
| | $\Phi_{\text{SOR},16}$ | 0.3757 | 0.3590 | 0.6168 | 0.7796 | 0.0309 | 0.0218 | 0.0224 |
| | $\Phi_{\text{SOR},64}$ | 1.2871 | 0.9696 | 0.4896 | 0.0872 | 0.3847 | 0.0492 | 0.0309 |

