# OpenReview forum: "Gradient-based Meta-solving and Its Applications to Iterative Methods for Solving Differential Equations"
_ICLR.cc/2022/Conference — ICLR 2022 Submitted_

### Official Review · Reviewer_ay6G · 2021-10-30

**Correctness:** 3
**Technical Novelty And Significance:** 2
**Empirical Novelty And Significance:** 2
**Recommendation:** 3
**Confidence:** 4

**Main Review:**

Strengths:
- The paper proposed to predict a good initial guess for traditional PDE solvers, so the PDE solver would converge fast. Also, by using traditional PDE solver, the obtained solution is usually more accuracy than other purely data-driven ML methods.

Weaknesses:
- The authors spent a lot of effort to create a new terminology “meta-solving”, which has a board meaning and many other algorithms can be formulated in this way. However, this is only a new terminology, but it is not a new idea or a new algorithm. From the paper, there is no clear evidence that why we would need this new terminology, or for what problems we have to use this new terminology.
- In fact, in the paper, it only tested the problem of generating good initial guess, which is not really a new idea.
- The paper only tested the algorithm on a 1D Poisson equation and 2D incompressible flow. Other challenging problems should be tested.
- There is no comparison between the proposed method and other methods in terms of inference speed and accuracy.
- The paper didn’t provide the details of networks used.


**Summary Of The Paper:**

The paper proposed a gradient-based algorithm GBMS to solve PDEs based on the solutions of other similar problems. In GBMS, a network is trained to produce good initial guess for the iterative solver of the PDE. Numerical experiments are performed to show the effectiveness of the method.

**Summary Of The Review:**

The paper introduced a new terminology and is more like a perspective paper, instead of a comprehensive research article.

---

> ### Author Response · Authors · 2021-11-18
> **Response to Reviewer ay6G**
>
> Thank you for the valuable comments. Regarding your concerns, we have provided our responses as follows.
>
> >The authors spent a lot of effort to create a new terminology “meta-solving”, which has a board meaning and many other algorithms can be formulated in this way. However, this is only a new terminology, but it is not a new idea or a new algorithm. From the paper, there is no clear evidence that why we would need this new terminology, or for what problems we have to use this new terminology.
> The paper introduced a new terminology and is more like a perspective paper, instead of a comprehensive research article.
>
> 1. The GBMS algorithm is to generalize a variety of existing meta-learning or similar approaches under the framework. It is not new by definition because it generalizes. For example, MAML, Feliu-Faba et al.  (2020), Psaros et al. (2021), and Chen et al. (2020) are all seen as special cases of this algorithm design pattern (section 2.3).
> 2. With respect to the application considered in this paper, the algorithmic novelties are as follows: a) Gradient-based meta-learning such as MAML gives a task-independent initialization, while the proposed method gives task-dependent initial guesses. b) Although several works, such as Huang et al. (2020), Ajuria Illarramendi et al. (2020), and Ozbay et al. (2021), propose to use task-dependent initial guesses generated by neural networks, these initial guesses are independent of the solvers. On the other hand, our initial guesses are optimized for each solver. Theorem1 and experiments demonstrate the importance of this difference.
> 3. We believe that this general framework is useful for the following reasons: 1) It enables us to understand the key differences between meta-learning approaches across machine learning and scientific computing applications. For example, MAML has a task-independent initialization, whereas the proposed method has task-dependent initial guesses. Other examples are discussed in section 2.3, and due to the unified framework, we can compare them regardless of their type of algorithms. 2) the general methodology gives us a common design pattern to develop numerical algorithms for new situations. For example, the GBMS differential equation solver proposed in this paper follows immediately from the formulation in section 2.1 and Algorithm1.
>
> >In fact, in the paper, it only tested the problem of generating good initial guess, which is not really a new idea.
>
> Although there are related works (Huang et al. (2020), Ajuria Illarramendi et al. (2020), and Ozbay et al. (2021)) on generating good initial guesses by neural networks, their choices are independent of the solver. On the other hand, our method depends on the solver, and Theorem1 and experiments show that the difference is important. These are remarked in 3rd paragraph of section 3.
>
> >The paper only tested the algorithm on a 1D Poisson equation and 2D incompressible flow. Other challenging problems should be tested.
>
> In this paper, we focused on developing the general framework and presented linear system solving as an example. We will study other types of problems such as nonlinear problems in future work.
>
> >There is no comparison between the proposed method and other methods in terms of inference speed and accuracy.
>
> In section 3.2, we added the comparison of actual computation time between our method and the traditional method. It shows that our method ($\Psi_{NN}$ with $\Phi_{SOR, 4}$) is approximately 10 times faster than the traditional one with better accuracy. We also compared with the supervised learning method ($\Psi_{NN}$ with $\Phi_{SOR, 0}$) that is similar to Ajuria Illarramendi et al. (2020). We observe that our method is approximately 6.6 times faster than the supervised learning baseline with better accuracy.
>
> >The paper didn’t provide the details of networks used.
>
> More detailed explanations about the networks are added in Appendix D.

---

> > ### Comment · Reviewer_ay6G · 2021-11-27
> > **Reviewer response**
> >
> > I have read other reviewers' comments and the discussion. The reviewers and the authors agree that the contribution of the paper is that it proposes a general framework---gradient-based meta solving, and some other algorithms can be formulated in such way. This sounds very "fancy", but the paper failed to show why it is useful for real applications. In the paper, it is only studied for linear system solving on generating good initial guesses (not really a new idea), which is OK but far away from enough.
> >
> > The main contribution of the paper is introducing a new terminology/framework, but it didn't really solve challenging problems. I hope to see new numerical experiments where the new framework can solve challenging problems and real applications. Also, because the framework is claimed to be general, then diverse applications of different types should be tested. Otherwise, the authors may also choose to rewrite the paper to be a perspective/opinion paper, even in the field of solving PDEs, then the authors don't have to perform new numerical experiments.

---

### Official Review · Reviewer_Wihm · 2021-11-02

**Correctness:** 3
**Technical Novelty And Significance:** 3
**Empirical Novelty And Significance:** 2
**Recommendation:** 8
**Confidence:** 3

**Main Review:**

Strengths:
- The paper proposes a very general formulation of “meta-solving” numerical problems. Thorough theoretical foundation and justifications are provided.

Weaknesses:
- The only use-case that is thoroughly empirical validated is solving PDEs. As the paper mentions, other applications, such as root-finding, are applicable. Only evaluating the framework on one application does not showcase its general applicability.
- Data augmentations are required for the incompressible flow simulation experiment. Why isn’t it possible for the meta-solver to learn without these augmentations?
- The formulation of the dataset for the experiment in 3.2 seems arbitrary. Why are the two previous timesteps required?

Are there stronger baselines that can be compared against? For example, are there problem-specific heuristic initial guesses that can be used that leverage domain knowledge about the particular problem?

Typos:
Section 2.2
- “is a algorithm”
- “to find a good initial weights”
- “for the meta-solving problems”
- “may not be an initial weights”
Section 2.3
- “\theta does not depend on task \tau” (paragraph 2)
- “\theta is weights of another” (paragraph 3)
- “In this work, meta-learning approach” (paragraph 4)
- “tested with multi steps of \Phi” (paragraph 4)
Algorithm 2
- “differntiable solver”
Section 3.1.2
- “tends to ignore high frequencies and more focus on low frequencies”  (paragraph 4)


**Summary Of The Paper:**

This paper proposes leveraging data from previous problem instances to improve efficiency of solving similar ones in the future. A general gradient-based method is proposed, which is applied to generating initial guesses to differential equation solutions. This problem is formulated as a meta-learning problem.


**Summary Of The Review:**

The paper proposes a general framework for efficiently finding solutions to numerical problems, but only evaluates the framework on PDE problems. Furthermore, additional tricks, such as data augmentations and using the previous two timesteps of the solution, are required to make the method work well empirically.

I’m not very familiar with meta-learning or PDE solvers, so I’m not very confident in my assessment.

---

> ### Author Response · Authors · 2021-11-18
> **Response to Reviewer Wihm**
>
> Thank you for the valuable comments. Regarding your concerns, we have provided our responses as follows.
>
> >The only use-case that is thoroughly empirical validated is solving PDEs. As the paper mentions, other applications, such as root-finding, are applicable. Only evaluating the framework on one application does not showcase its general applicability.
>
> In this paper, we focused on developing the general framework and presented linear system solving as an example. We will study other types of problems in future work.
>
> >Data augmentations are required for the incompressible flow simulation experiment. Why isn’t it possible for the meta-solver to learn without these augmentations?
>
> Without the data augmentation, the inputs of the neural network are from the prepared dataset and always accurate. However, during inference, the inputs of the neural network are the outputs of the simulation at previous timesteps, which can be a slightly different distribution than encountered during training (i.e. distribution shift). This then causes further accuracy problems for the simulation at the next step, and this issue compounds itself in time. This eventually degrades the performance if the shift in distribution is not dealt with. The data augmentation is used to prevent it. Note that this problem is specific to cases where the meta-solver interacts with an outer-loop solver, which controls the task distribution. We update the explanation about the data augmentation in section 3.2.
>
> >The formulation of the dataset for the experiment in 3.2 seems arbitrary. Why are the two previous timesteps required?
>
> Although $p_\tau$ is completely determined by $f_\tau$ theoretically, previous timesteps can provide useful information to determine a good guess for $p_\tau$. For example, $p_\tau$ should not be so far from $p_{\tau-1}$, and $p_{\tau-1} - p_{\tau-2}$ can tell whether the pressure is increasing or decreasing, and how fast. In fact, these additional features improve the performance. We added the explanation in section 3.2.
>
> >Are there stronger baselines that can be compared against? For example, are there problem-specific heuristic initial guesses that can be used that leverage domain knowledge about the particular problem?
>
> We would say that using the previous timestep pressure for the incompressible flow simulation is one of the problem-specific heuristic initial guesses based on the domain knowledge. In the literature of computational fluid dynamics (CFD), researchers focus on how to design an efficient preconditioner rather than the choice of initial guess because the choice of preconditioners usually incorporates prior knowledge easily - an example would be to take the diagonal (or tri-diagonal) approximation of the differential operator in the NS equations. Although some methods are proposed to choose a good initial guess (e.g. Ye et al. (2020)), to the best of our knowledge, the standard choice in practice is the previous timestep pressure.
>
> >Typos
>
> Thank you for pointing out the typos. They are fixed.
>
> Reference:
>
> Ye, S., Lin, Y., Xu, L., & Wu, J. (2020). Improving Initial Guess for the Iterative Solution of Linear Equation Systems in Incompressible Flow. Science in China, Series A: Mathematics, 8(1), 119.

---

> > ### Comment · Reviewer_Wihm · 2021-11-21
> > **Response**
> >
> > Thanks for your response. I think you've addressed all of my concerns. I will increase my score, but AC please keep in mind that I am not very familiar with this area, so I do not have a high confidence in my assessment.

---

### Official Review · Reviewer_ExFG · 2021-11-02

**Correctness:** 4
**Technical Novelty And Significance:** 2
**Empirical Novelty And Significance:** 2
**Recommendation:** 5
**Confidence:** 4

**Main Review:**

Strengths:
1. The paper applies gradient-based methods to important problems of learning initializers for iterative methods in scientific computing.
2. The authors provide a guarantee for initializing the Jacobi iteration, albeit under what seems like a restrictive assumption on the model capacity.
3. The authors demonstrate improvement over simply learning one mapping (“supervised learning”) rather than back propagating through iterations.

Weaknesses:
1. While the improvement over the “supervised learning” setting is interesting, the evaluation largely seems to be in regimes where the error is far too high for practical applications. For example, in Table 1 the MSE of even the best approach seems quite high, although it is difficult for me to get a sense of what a good scale is. Would the advantage continue to hold and be significant-enough to be interesting if the methods were given sufficient number of iterations for practical purposes?
2. There is no demonstration of practical application utility, i.e. whether going through the trouble learning this initialization is actually useful. Is it more useful for me to spend the (likely substantial) amount of effort of back propagating through a lot of classical solves in order to get a better initialization, or just to use the classical solver to begin with. As an example, in the field of neural PDE solvers there is often a demonstration of end-to-end computational savings provided (c.f. Li et al., (2021)).
3. While the claimed framework is very general, it is only studied for linear system solving. The authors also do not compare their overall framework to the substantial work on data-driven algorithm design, which has been studying these problems both theoretically and empirically for quite some time (e.g. Hutter et al. (2011), Balcan (2020), Mitzenmacher & Vassilvitskii (2020)).

References:
Balcan. *Data-driven Algorithm Design*. In Roughgarden, *Beyond the Worst-Case Analysis of Algorithms,* 2020.
Hutter et al. *Sequential model-based optimization for general algorithm configuration*. ICLLO 2011.
Li et al. *Fourier Neural Operator for Parametric Partial Differential Equations*. ICLR 2021.
Mitzenmacher & Vassilvitskii. *Algorithms with Prediction*. In Roughgarden, *Beyond the Worst-Case Analysis of Algorithms,* 2020.

**Summary Of The Paper:**

This paper introduces a framework for learning the parameters of computational algorithms in order to improve runtime and/or error. They study the specific case of linear systems that arise in PDE solvers, showing an objective whose solution is an initialization that decreases the number of Jacobi iterations required to solve the Poissson equation as well as empirical results for both Jacobi and SOR on several PDE systems.

**Summary Of The Review:**

While the problem setup is reasonably well-motivated and some of the empirical results are interesting, it is not clear to me how practically relevant the empirical results are for the problems being studied. The very general framework is also only discussed in the restricted case of linear system solving for PDEs. As a result I tend to lean against acceptance.

---

> ### Author Response · Authors · 2021-11-18
> **Response to Reviewer ExFG**
>
> Thank you for the valuable comments. Regarding your concerns, we have provided our responses as follows.
>
> >While the improvement over the “supervised learning” setting is interesting, the evaluation largely seems to be in regimes where the error is far too high for practical applications. For example, in Table 1 the MSE of even the best approach seems quite high, although it is difficult for me to get a sense of what a good scale is. Would the advantage continue to hold and be significant-enough to be interesting if the methods were given sufficient number of iterations for practical purposes?
>
> The MSE at 10,000 iterations is provided in Figure2, and $\Psi_{NN}$ trained with $\Phi_{SOR, 64}$ gives a relative error of 0.013 at 10,000 iterations. This remains approximately 8% better than $\Psi_{NN}$ trained with $\Phi_{SOR, 0}$ (supervised learning). Thus, the proposed method is still effective for a large number of iterations for practical purposes. We added this explanation in section 3.1.2.
>
> >There is no demonstration of practical application utility, i.e. whether going through the trouble learning this initialization is actually useful. Is it more useful for me to spend the (likely substantial) amount of effort of back propagating through a lot of classical solves in order to get a better initialization, or just to use the classical solver to begin with. As an example, in the field of neural PDE solvers there is often a demonstration of end-to-end computational savings provided (c.f. Li et al., (2021)).
>
> The incompressible flow simulation is a demonstration of practical utility of our method. In the incompressible flow application, it takes 11 sec for the trained meta-solver and 4 SOR iterations to simulate the flow for 1 second. On the other hand, it takes 105 sec for the heuristic initial guess and 128 SOR iterations, whose error is 0.001431 and still worse than the trained meta-solver and 4 SOR iterations. Because the training time is approximately 7 hours, our approach is effective if we simulate the flow for more than 268 seconds, which can be easily satisfied in practical settings. We added the comparison in section 3.2.
>
> >While the claimed framework is very general, it is only studied for linear system solving.
>
> In this paper, we focused on developing the general framework and presented linear system solving as an example. We will study other types of problems such as nonlinear problems in future work.
>
> >The authors also do not compare their overall framework to the substantial work on data-driven algorithm design, which has been studying these problems both theoretically and empirically for quite some time (e.g. Hutter et al. (2011), Balcan (2020), Mitzenmacher & Vassilvitskii (2020)).
>
> Thank you for the valuable references. The difference between the work on data-driven algorithm design and our framework is that the former focuses on discontinuous problems such as combinatorial optimizations while the latter focuses on differentiable ones. Hence, these algorithms cannot directly be applied to the problems studied in this paper. We added the suggested references and further explanation on this point in section 2.2 in revision. Also, we have made comparisons with baseline methods (both classical and data-driven) in this paper. For example, in section 3.12, the classical baseline $\Psi_{BL}$ is the previous timestep pressure, and the data-driven baseline is $\Psi_{NN}$ with $\Phi_{SOR, 0}$ that is similar to Ajuria Illarramendi et al. (2020).

---

> > ### Comment · Reviewer_ExFG · 2021-11-22
> > **Response**
> >
> > Thank you for the response.
> >
> > 1. 8% improvement seems rather small, and the advantage seems to shrink dramatically with more iterations. Furthermore, even 10^-2 error seems rather large from my (admittedly somewhat dated) experience with PDE solvers.
> >
> > 2. Thanks for the clarification. I agree that this is a demonstration of usefulness, although in light of my concerns with the rather high relative errors being studied (discussed in point 1) it is unclear to me that the improvement is significant in settings where such low-precision solves would not be acceptable.
> >
> > 3. I think it is more accurate to describe the contribution here as a type of data-driven algorithm design, as it studies both discrete and continuous problems.

---

### Official Review · Reviewer_yRZK · 2021-11-04

**Correctness:** 4
**Technical Novelty And Significance:** 3
**Empirical Novelty And Significance:** 3
**Recommendation:** 6
**Confidence:** 3

**Main Review:**

The problem is well formulated. Numerical integrators of differential equations can be sensitive to choices of parameters or initial conditions. At the same time numerical integration is a computationally complex task that can benefit from meta-learning. Iterative solvers and their surrogate meta-solvers can both be implemented by a neural networks which makes the problem approachable by gradient based meta learning.

The authors demonstrate their approach on a family of 1D Poisson equations and incompressible flow simulations. The authors successfully demonstrate the advantages of using gradient based meta solving with a neural network architecture for the meta-solver over a baseline learner and over regular supervised learning on this task.

**Summary Of The Paper:**

The authors define the task of solving a family of differential equations as a task of gradient-based meta-learning generalizing the gradient-based model agnostic meta-learning to problems with differentiable solvers.

**Summary Of The Review:**

The presented work is solid. The main concern is with its limited audience in the scope of the conference and potential applications beyond those presented in the paper.

---

> ### Author Response · Authors · 2021-11-18
> **Response to Reviewer yRZK**
>
> Thank you for the valuable comments. Regarding your concerns, we have provided our responses as follows.
>
> >The main concern is with its limited audience in the scope of the conference and potential applications beyond those presented in the paper.
>
> The meta-solving framework includes not only solving differential equations, but also other numerical algorithms. In the paper, we showed that MAML (Example 1), learning-based methods including both meta-learning and regular supervised learning (Section 2.3), and our proposed method can be formulated under the same framework. Thus, we expect that the audience in ICLR can find interest and applications of our paper.

---

> > ### Comment · Reviewer_yRZK · 2021-11-18
> > **Scope**
> >
> > Thank you for the response and clarification. True, you do give an example of the potential use of the proposed approach in a meta-learning and "regular" supervised learning. However, these claims and examples are not supported by the results shown in the paper. The paper, as the title suggests, focuses on its application to solving differential equations. This application, as I stated before, has merit, but might be a niche that resonates with a limited ICLR audience. Making a better connection in the text to the other possible applications, during the review period might be possible and will be an improvement. However, making the claim and demonstrating a broader scope might be possible, would require significant experimental evaluation and comparison.

---

### Decision · Program_Chairs · 2022-01-20

**Decision:**

Reject

**Comment:**

The authors define the task of solving a family of differential equations as a task of gradient-based meta-learning generalizing the gradient-based model agnostic meta-learning to problems with differentiable solvers. According to the reviews, there were some concerns regarding the practical value of the paper, for example, (1) the proposed technology is restricted to linear systems, and relatively easy problems (2) there is no demonstration of practical application utility (3) It lacks systematic comparison with other methods (4) some technical details are missing. There were quite a lot of discussions on the paper among the reviewers, and the consensus is that the paper is not solid enough for publication at ICLR in its current form (the reviewer who gave the highest score is less confident and does not want to champion the paper).